# Effect of Terahertz Electromagnetic Field on the Permeability of Potassium Channel Kv1.2

**DOI:** 10.3390/ijms241210271

**Published:** 2023-06-17

**Authors:** Wen Ding, Xiaofei Zhao, Hongguang Wang, Yize Wang, Yanjiang Liu, Lirong Gong, Shu Lin, Chunliang Liu, Yongdong Li

**Affiliations:** 1Key Laboratory for Physical Electronics and Devices of the Ministry of Education, Xi’an Jiaotong University, Xi’an 710049, China; dingwen@mail.xjtu.edu.cn (W.D.); zhaoxiaofei@stu.xjtu.edu.cn (X.Z.); wanghg@mail.xjtu.edu.cn (H.W.); wangyize@stu.xjtu.edu.cn (Y.W.); yanjiang@stu.xjtu.edu.cn (Y.L.); gonglr@stu.xjtu.edu.cn (L.G.); shulin@mail.xjtu.edu.cn (S.L.); 2School of Electronic Science and Engineering, Xi’an Jiaotong University, Xi’an 710049, China

**Keywords:** Kv1.2 ion channel, ion permeation mechanism, terahertz electromagnetic field, hydrogen bond lifetime

## Abstract

In this paper, the influence of external terahertz electromagnetic fields with different frequencies of 4 THz, 10 THz, 15 THz, and 20 THz on the permeability of the Kv1.2 voltage-gated potassium ion channel on the nerve cell membrane was studied using the combined model of the “Constant Electric Field-Ion Imbalance” method by molecular dynamics. We found that although the applied terahertz electric field does not produce strong resonance with the –C=O groups of the conservative sequence T-V-G-Y-G amino acid residue of the selective filter (SF) of the channel, it would affect the stability of the electrostatic bond between potassium ions and the carbonyl group of T-V-G-Y-G of SF, and it would affect the stability of the hydrogen bond between water molecules and oxygen atoms of the hydroxyl group of the 374THR side chain at the SF entrance, changing the potential and occupied states of ions in the SF and the occurrence probability of the permeation mode of ions and resulting in the change in the permeability of the channel. Compared with no external electric field, when the external electric field with 15 THz frequency is applied, the lifetime of the hydrogen bond is reduced by 29%, the probability of the “soft knock on” mode is decreased by 46.9%, and the ion flux of the channel is activated by 67.7%. Our research results support the view that compared to “direct knock-on”, “soft knock-on” is a slower permeation mode.

## 1. Introduction

Many physiological processes of organisms are closely related to the permeability of ion channels on the cell membrane, such as the maintenance of cell membrane potential, the generation of action potentials, the conduction of nerve signals, the regulation of the central nervous system, heart beating, skeletal muscle contraction, hormone secretion, and so on [1]. Abnormal ion channel permeability usually leads to a variety of diseases, such as cardiovascular and cerebrovascular diseases, neurodegenerative diseases, tumors, etc. These diseases, also known as ion channel defect diseases, are generally considered to be genetic diseases and incurable [1,2,3].

Modern medical research believes that the vibration and rotation frequencies of deoxyribonucleic acid (DNA), ribonucleic acid (RNA), and protein macromolecules in living organisms are in the frequency range of terahertz electromagnetic field, and the quantum energy of terahertz wave is much lower than the valence electron ionization energy of biological molecules (typical value is several eV) [4,5,6,7,8]. Therefore, it will not directly destroy biomolecules but can affect their conformation and function through nonlinear resonance. This special biological effect may realize the regulation of ion channel permeability [9], which is expected to be used in the treatment of ion channel disease.

In recent years, some studies have proved that external terahertz electromagnetic fields can change the permeability of ion channels. For example, Liu Xiaming’s team proved that terahertz electromagnetic fields (0.1–3 THz) could change the permeability of calcium ion channels and thus affect sperm motility through the experiment of irradiation of sperm droplets by terahertz waves, significantly increasing the forward motion rate of sperm [10]. Tan Xiaoxuan et al. found that a 34.88 THz electromagnetic field resonates with the binding sites (–C=O groups) on SF of the potassium channel in the nerve membrane of the cochlea, increasing K+ permeability and improving cochlea hearing [11]. The study by Liu Xi et al. showed that a 53.7 THz electromagnetic field could induce resonance of the –C=O on SF of the potassium ion channel and increase the permeability of potassium ions [12]. Li Yangmei et al. proved that the permeability of the calcium ion channel could be increased under the stimulation of the 42.55 THz electromagnetic field [13]. Chang Chao’s team adopted molecular dynamics simulation and found that under an electromagnetic field with an amplitude of 2.5 V/nm and frequency of 1.39 THz, the hydrogen bond of one-dimensional water molecules in the bionic channel was broken into a single water molecule, resulting in the phenomenon of super-penetration. The water permeability shows exponential growth [14]. L Guo et al. found that the applied electric field of 1 THz can improve the permeability of Ca^2+^ in calcium ion channels through the Brown dynamics (BD) physical model [15]. Zhang et al. demonstrated through molecular dynamics simulation that a 0.1 THz terahertz electric field can increase the channel current of the KcsA ion channel by 4 times [16]. Sun Kunyuan et al. found that under the action of the 53.7 THz terahertz electromagnetic field, the number of α helix in KcsA potassium channel protein decreased, the number of β folding and curling increased, and the permeability of potassium ions increased [17]. Yize Wang et al. found that a 51.87 THz electric field greatly improves KcsA channel permeation [18].

The above studies indicate that terahertz electromagnetic fields may interact with the SF of the ion channel [11,12,18] or with potassium ions in the channel [16]. Some studies even show that terahertz electromagnetic fields can change the secondary structure of ion channels [17]. All the effects change the permeability of ion channels. These studies are preliminary. The mechanism of potassium channels is complex and controversial. Much research needs to be conducted to find out the nature of the interaction between terahertz electromagnetic fields and ion channels and the possible application of such interactions.

In this paper, the influence of external terahertz electromagnetic fields on the permeability of the Kv1.2 potassium ion channel was studied. Kv1.2 is a typical voltage-gated potassium channel located on the nerve membrane. It is the basic excitation unit of the nervous system and plays an important role in the generation and conduction of electrical signals. In our study, the frequencies of external electric fields are 4 Hz, 10 Hz, 15 Hz, and 20 Hz. It is a new attempt to explore the interaction mechanism between the external terahertz field and the ion channel.

## 2. Results and Discussion

### 2.1. Effect of External Terahertz Electric Field on Ion Channel Current

Under physiological conditions, when a potassium ion goes from the cavity to the SF region, it should remove the hydration molecules first. Because the SF is the narrowest part of the channel with a radius of about 1.4 Å, the radius of the hydrated potassium ion is greater than the radius of the SF region, and each K^+^ potassium ion (Pauling radius 1.33 Å) must shed its hydration molecules to enter the SF region one by one. When the ion enters the SF region, it will pass through each binding site of S_cav_-S_4_-S_3_-S_2_-S_1_-S_0_ successively and finally exit the SF. These sites are key to ensuring both high selectivity and high conductance for K^+^ [19]. So, most studies of the effect of applied electric fields on ion channel permeability have focused on frequencies in the infrared region that can strongly resonate with SF carbonyl groups [11,12,13,14,15,16,17]. We consider the special bioelectromagnetic effect of terahertz electric fields, as well as the complexity of potassium channel permeability mechanisms. Here, we investigate the effect of electric fields with frequencies outside the infrared region on the permeability of ion channels. The applied electric field with a frequency less than or equal to 20 THz was selected for the study. The frequencies were 4 THz, 10 THz, 15 THz, and 20 THz.

The simulation results are shown in Figure 1 and Table 1. The model parameters and calculation details are described in the final section. We conducted six independent repeated simulations at each condition, and each simulation lasted 400 ns, giving a total simulation duration of 2.4 µs for each condition.

Figure 1a is the root mean square deviation (RMSD) of the channel during the first production simulation for each condition, which is used to judge whether the system is stable. We can see that regardless of whether there is an external electric field, the RMSD fluctuation is very small, which is between 0.15 and 0.35 at each condition, indicating that the structure of the channel is relatively stable during the simulation for each condition. In the Appendix A (Appendix A), we show six RMSD curves for each condition. It can be seen that each independent simulation has a similar variation trend and maximum value.

Figure 1b shows the molecular dynamics simulation results for ion fluxes. The ion current and channel conductance are obtained based on the ion flux, and the results are summarized in Table 1. Under physiological conditions, the potassium ion transport rate is 10^8^/s, and our simulation results are close to it, indicating that the simulation results are reliable.

According to Figure 1b and Table 1, an external electric field with different frequencies increases the ion flux, ion current, and ion conductance to a different extent compared to the case without an electric field. Especially when the applied electric field frequency is 15 THz, the maximum value is reached, and the ion flux increases by 67.7%.

### 2.2. Effect of External Terahertz Electric Field on Ion Permeation Mode

It has been found that potassium ions have their unique law for passing through the SF region. It is known as the permeation mode [20,21,22]. Here, we investigate the ions’ movement trajectory and permeation mode as they pass through the SF region with or without an external terahertz electric field.

The positively charged potassium ion would be attracted by the negatively charged oxygen atom of carbonyl –C=O on the conserved T-V-G-Y-G amino acid residue at each binding site, forming the electrostatic bond. So, potassium ions must break electrostatic binding at each binding site to reach other sites [20,21,22]. Figure 2 shows the trajectory of potassium ions passing through the SF region, and the color curve represents the trajectory of each potassium ion along the selectivity filter. The trajectory of potassium ions along the selectivity filter is tortuous. This indicates that the potassium ion vibrates at each binding site until it can cross the energy barrier before leaving the binding site.

Figure 3 shows the residence time of potassium ions at each binding site under different conditions. We can see that an applied electric field can shorten the vibration time of ions at each binding site. The electric field of 15 THz reduced the vibration time of ions at each binding site most significantly.

The permeation mode of ions passing through the SF region can be obtained according to the snapshots of ion position changes in the SF region at different times, and the results are shown in Figure 4. There are three kinds of ion permeation modes when potassium ions pass through the SF region: “soft knock-on” mode, “direct (hard) knock-on “mode, and “mixed knock-on” mode.

The “soft knock-on” mode is shown in Figure 4a. Ions pass through the SF with water molecules in the pattern of “K^+^-W-K^+^-W” and “W-K^+^-W-K^+^” (W stands for water molecule). It can be seen that the SF consists of two resident K^+^ separated by a water molecule. The “soft knock-on” mode was found by MacKinnon and his colleagues in 2003 and proposed by Hummer [21]. This ion permeation mechanism assumes that water molecules have a synergistic effect to reduce the dielectric constant and increase the ion transfer rate. Water molecules are essential to the ion transfer process. This view is still widely held today.

The “direct knock-on” mode is shown in Figure 4b. Ions pass through the SF in a pattern such as “K^+^-O-K^+^-K^+^” and “K^+^-K^+^-O-K^+^” (O is for a vacancy). It can be seen that there is no water molecule between the ions when they pass through the SF region. This different permeation mechanism was proposed by Kopfer et al. in 2014. It assumes that the key to efficient conductivity of K^+^ lies in the direct Coulomb collision between adjacent ions rather than the water molecules co-permeate with K^+^ [22,23].

The “mixed knock-on” mode is shown in Figure 4c. It can be seen that the ions pass through the SF in the pattern as “K^+^-W-K^+^-K^+^” or “K^+^-K^+^-W-K^+^”. There is only one water molecule through the SF. It is more like a transition mode between the “soft” and “direct” knock-on modes.

The occurrence probability of the three modes was counted, and the results are shown in Figure 4d. Compared with the case without an electric field, we can see that the external electric field of different frequencies changes the occurrence probability of the three modes. When an external electric field is present, the probability of “soft knock-on” mode is reduced, while “direct knock-on” and “mixed knock-on” modes are increased. Especially when the frequency is 15 THz, the “soft knock-on” mode occurrence probability is reduced by 46.9%, while the “direct knock-on” mode occurrence rate is increased by 45.5%.

Since the ion flux is highest at 15 THz compared with other cases (Figure 1b), our results are consistent with Kopfer’s theory [22,23], indirectly proving that the “soft knock-on” mode is a slower permeation mode for potassium ion channels. Water molecules are not essential for the permeation of potassium ions.

### 2.3. Analysis of the Influence of External Terahertz Electric Field on the Permeability of Potassium Ion

In order to explain the influence of an external terahertz electric field on potassium ion permeability, the following analysis was performed.

#### 2.3.1. Effect of Terahertz Electric Field on –C=O Groups on the TVGYG Amino Acid Residue of SF

The infrared absorption spectrum of the –C=O groups on the TVGYG amino acid residue at sites S_0_–S_4_ in the SF region was calculated by GROMACS software [24], and the results are shown in Figure 5a. It can be seen from the spectrum diagram that the main peaks of carbonyl vibration in SF are after 30 THz; the carbonyl groups of the SF do not have strong absorption peaks at 4 THz, 10 THz, 15 THz, and 20 THz.

Figure 5c shows the pore diameter of the SF over time. The pore diameter of the SF is the distance between the carbonyl oxygen atom on each residue and the carbonyl oxygen atom on the symmetric residue in the SF. The change in SF average pore diameter under the action of different frequency electric fields is shown in Figure 5b. It can be seen that the SF pore diameter is stable under different electric fields except for Y377. The aperture at Y377 increases when the external field frequency is 4 THz or 20 THz. The results show that the applied terahertz electric field does not resonate with the SF binding site –C=O. This means it could not strengthen the vibration or twist of the –C=O bond to induce a conformational change in the SF. From Figure 5b, we can conclude that the applied electric field with a frequency of 15 THz has little effect on the average SF pore diameter compared to no field. Therefore, the change in ions permeability cannot be caused by the SF conformational change.

The positively charged potassium ions are attracted to the negatively charged polar carbonyl group of SF’s conserved T-V-G-Y-G amino acid residues, forming electrostatic bonds [20,21,22]. So, potassium ions must overcome the barriers between binding sites to pass through each binding site. If the external terahertz electric fields make the electrostatic bond unstable and easy to break, it will also affect the permeability of the potassium ions. Therefore, we analyze the effect of an applied electric field on ions’ mean force potential (PMF) in the next section.

#### 2.3.2. Effect of Terahertz Electric Field on Potential of Mean Force (PMF)

The mean force potential (PMF) is the negative logarithmic of K^+^ ions’ densities in the SF. The logarithmic of ion densities represents the quasi-free energies of the ions [22,25,26]. Thus, the PMF represents the potassium ion density and free potential at the binding site.

Figure 6a is the PMF distribution in SF as the K^+^ ions pass through the SF region with or without external terahertz electric fields. The dotted lines in Figure 6a are the positions of the oxygen atoms of the carbonyl groups, and between the two dotted lines are the positions of the canonical potassium binding sites. The potential minima (density maxima) reflect stable bound ions. It is important to note that the higher the potential value, the more unstable the potassium ion in that position is.

According to Figure 6a, the changes in ions’ PMF at each binding site under different conditions show that the applied electric field does alter the stability of the electrostatic bond between potassium ions and carbonyl groups. The preference of each site for potassium ions varies under different electric field conditions. The occupancy rate of potassium ions at each site is shown in Figure 6b, and it can be seen that an applied terahertz electric field could change the rate of potassium ions in the SF. It can be verified more directly that an applied terahertz electric field could affect the potential energy distribution of potassium ions in the SF, which is expressed as a change in the affinity of sites for potassium ions.

Figure 6b can also help us understand why the shape of the potential curve of potassium ions in Figure 6a differs for each condition. For example, in Figure 6a, potassium ions can form a shallow potential well at the S_1_ site without applying fields, which is absent in other cases. This is because the site occupancy of potassium ions at S_1_ is significantly higher without applying fields than in other cases. Without applying fields, we speculate that potassium ions will stay at the S_1_ site for a while when they translate from S_2_ to S_0_. However, a direct path from S_2_ to S_0_, without staying in S_1_, might be more prevalent in other circumstances. This is also related to the differences in permeation modes under different conditions, as we found by tracing the potassium ion trajectories that the “direct knock-on” (K^+^-K^+^-O-K^+^) mode and “mixed knock on” (K^+^-K^+^-W-K^+^/K^+^-W-K^+^-K^+^) mode, the probability of ions appearing in S_1_ is the lowest. However, in the “soft knock-on” (K^+^-W-K^+^-W) mode, the probability of potassium ions appearing at S_1_ is almost the same as it appeared at other sites. We have discussed earlier that soft knock-on is a slower permeation mode for potassium ions. This further suggests that the permeation path of S_2_-S_1_-S_0_ is slower than the direct path from S_2_ to S_0_. Therefore, although the potential barrier of potassium ions at the S_1_ site in the absence of applied fields is low, it does not mean potassium ions can pass through the SF more quickly.

From the above analysis, we found that it is difficult to compare the heights of PMFs directly. This is because it involves different site occupancy states and permeation modes. However, we believe that the depth of the potential well at the S_cav_ site is crucial, because as long as the potassium ion at the S_cav_ site enters the SF, regardless of the occupancy state in the SF, it always forces the existing potassium ion in the SF to move outside the membrane to complete the transmembrane motion. As can be seen in Figure 6a, the minimum value of the potential well at 15 THz is the largest, which means that potassium ions can enter the channel faster. This is consistent with the conclusion derived from the residence time of potassium ions shown in Figure 3. As a result, the electric field at 15 THz is favorable for improving the ion flux.

#### 2.3.3. Effect of Terahertz Electric Field on the Hydrogen Bond Formed by Water Molecules and Side Chain Oxygen Atoms of Threonine Residues (374THR) at the SF Entrance

We know the existing permeation mechanism of potassium ions is the “soft knock-on” and “direct knock-on” permeation mode. The dispute between them lies in whether water molecules are required for the efficient permeation of potassium ions [22]. In the physiological situation, the oxygen atoms of hydroxyl of 374THR (S_c_ site) are surrounded by water molecules and form hydrogen bonds with them. When the hydrogen bond is broken, the water molecule could move deeper into the SF or return to the cavity. If the water molecule moves deeper into the SF, the “soft knock-on “mode is formed; if the water molecule goes back to the cavity, the “direct knock-on” mode is formed. Figure 7a shows the difference in water molecules’ movement in two modes. We can know that it is necessary to break the hydrogen bond between water molecules and the oxygen atom of the hydroxyl side chain of 374THR first, regardless of the kind of permeation mode. So, we analyzed the external electric fields’ effects on the stability of the hydrogen bond.

The lifetime of the hydrogen bond between water molecules and the oxygen atoms of the hydroxyl of 374THR under different conditions was calculated, as shown in Figure 7b. It can be seen that the external electric fields shorten the lifetime of the hydrogen bond, with the most significant reduction (29%) at 15 THz. The shorter lifetime of the hydrogen bond means that the external electric field makes the hydrogen bond unstable, increasing the probability that bound water molecules become free. Furthermore, it is easier for them to return to the cavity than go deeper into the SF. As a result, the potassium ions could enter SF more easily. We speculate that the stability of hydrogen bonding is also related to the motion pattern of potassium ions in SF because, comparing Figure 7b with Figure 4d, we can know that the lowest hydrogen bonding lifetime with a 15 THz electric field corresponds to the lowest soft knock-on mode ratio and the highest direct knock-on mode ratio.

## 3. Materials and Methods

### 3.1. Simulation Model and Method

The model is based on the open-state crystal structure of Kv1.2 (PDB ID: 3lut), a voltage-gated potassium ion channel [19]. It is a tetramer structure consisting of four subunits with a pore domain (PD) in the center and a voltage sensor domain (VSD) around it. The VSD responds to the change in membrane potential and controls the opening and closing of the PD. The PD controls the permeation of ions. The PD can be divided into three regions from intracellular to extracellular: the gate, the cavity, and the selectivity filter (SF), as shown in Figure 8a. The SF consists of a conserved sequence T-V-G-Y-G, as shown in Figure 8b. The oxygen atoms of carbonyl groups or the hydroxyl group of the conserved sequence T-V-G-Y-G face the inner center form binding sites: S_0_, S_1_, S_2_, S_3_, S_4_, and S_c_ sites. Binding sites provide a tight binding for K+ ions when they pass through the SF. It has been found that potassium ions have specific permeation modes at SF binding sites, and these permeation modes are responsible for the high speed of potassium ion transport [20,21,22].

By far, the constant electric field (CEF) method [27] is the main method of molecular dynamics (MD) used to simulate the effect of an external electric field on ion channels. The CEF method originates from the Poisson−Boltzmann theory. Based on the theory, the transmembrane potential is represented by an external electric field applied along the normal direction of the membrane. Thus, the effect of the applied electric field on the protein channel can be obtained by changing the electric field normally along the membrane. The method is effective, but it is quite different from the actual situation. This is because the transmembrane potential of the ion channel is caused by the ion concentration gradient under physiological conditions. The ion imbalance (IIMB) method [25] is a method that is closer to physiological conditions. The transmembrane potential is generated by a small charge imbalance Δq that is produced by exchanging ions on both sides of the membrane. The Δq can be sustained by a forced recovery by exchanging ions on one side of the cell membrane and water molecules on the other side in the method. Combining these two methods, we propose a joint model: the “Constant Electric Field—Ion Imbalance” combined model. In this model, the transmembrane potential is produced by a small charge imbalance Δq of the IIMB method, and the external terahertz electromagnetic field on the channel comes from the CEF method. The combined model is shown in Figure 9.

The applied terahertz electromagnetic field can be treated as a quasi-static terahertz electromagnetic field in the ion imbalance–constant electric field combined model.

The wavelength of a 10 THz electromagnetic wave is 30 μm, and the size of the molecular dynamics model is usually nanoscale. The wavelength of a terahertz electromagnetic wave is much larger than the simulation geometric dimensions. The applied terahertz electromagnetic field can be considered quasi-static. It can be treated as an electric field and a magnetic field varying with time separately, without considering the wave property.

When a particle with charge *q* and velocity *ν* meets a planar electromagnetic wave with the electric field intensity *E*(*t*) and magnetic field intensity *H*(*t*) in free space, it would subject the electric field force $f_{e}$ and magnetic field force $f_{m}$_,_ as below:(1)$$f_{e}=qE(t)$$
(2)$$f_{m}=qv\mu_{0}H(t)$$
(3)$$\frac{f_{m}}{f_{e}}=v\mu_{0}\frac{H(t)}{E(t)}=v\mu_{0}\sqrt{\frac{\varepsilon_{0}}{\mu_{0}}}=v\sqrt{\mu_{0}\varepsilon_{0}}=\frac{v}{c}$$

The particle’s velocity ν is far less than the light speed *c*, so the value of $f_{m}/f_{e}$ is very small, which means the electric force $f_{e}$ of the particle is much greater than the magnetic force $f_{m}$. Only the terahertz electric field was considered in the molecular dynamic simulation. So, the applied terahertz electromagnetic field can be treated as a quasi-static terahertz electric field in the ion imbalance–constant electric field combined model. It can be expressed as the following formula:(4)$$E(t)=E_{0}\cos[\omega (t-t_{0})]$$

Here, the *E*_0_ is the electric field intensity, and ω is the angular frequency.

### 3.2. Modeling Details

The model consists of a Kv1.2 ion channel and palmitoyl-oleoyl-phosphatidylcholine (POPC) membrane, with the ion channel embedded in a 9 nm × 9 nm POPC membrane. The system was set up by the CHARMM-GUI web server [28], which contains 12,131 TIP3P water molecules, 213 potassium ions, and 205 chloride ions. Three potassium ions were put into the selectivity filter, and one was put in the cavity to perform the original model. The bi-layer membrane system was established for the use of CompEL (IMB) [25], and the charge difference on the two sides of the membrane was 4e. The applied electric field was established by the CEF method [27], with an amplitude of 0.4 V/nm and frequency of 4 THz, 10 THz, 15 THz, and 20 THz. The whole system is shown in Figure 9. During the production simulation, no restrictions were applied.

### 3.3. Simulation Settings

MD simulations were performed using GROMACS (2019.4) software with a CHARMM36 force field [29,30,31]. Six production simulations of 400 ns were performed for each condition. Linear constraint solver (LINS) was used to restrict covalent bonds containing hydrogen atoms. The time step was set to 2 fs. The Verlet list scheme was used to update the nearest neighbor tables. The particle mesh Ewald (PME) method was used to deal with electrostatic force. The cutoff distance was set at 1.2 nm. The system was kept at 310 K and 1 atm by a V-rescale thermostat and Parrinello–Rahman barostat.

### 3.4. Data Analysis Method

The main data analysis methods in the paper are as follows.

Residence time calculation for potassium ion: The S_0_ site region was defined between the position of the carbonyl oxygen atom of 378GLY and the carbonyl oxygen atom of 377TYR. We calculated the average residence time of potassium ions at the S_0_ site by dividing the total time the potassium ions stayed in the region by the total number of ions passing through the channel. Similarly, the position of the 377TYR and 376GLY amino acid carbonyl oxygen atoms was used to define the S_1_ region; the position of 376GLY and 375VAL amino acid carbonyl oxygen atoms was used to define the S_2_ region; the position of 375VAL and 374THR amino acid carbonyl oxygen atoms was used to define the S_3_ region; the position of the 374THR amino acid carbonyl oxygen atom and the 374THR hydroxyl oxygen atom was used to define the S_4_ region; and the S_cav_ site was defined as within 0.3 nm down from the 374THR side chain hydroxyl oxygen atom. The average residence times of ions at S_1_, S_2_, S_3_, S_4_, and S_cav_ were obtained sequentially by the same method as that used to calculate the average residence time of S_0_.

Loading state in the SF: The loading state in SF is visualized by VMD software. When the potassium ions in the SF leave the channel, the previous frame’s loading state is recorded. If the loading state in the channel is KKK, it is recorded as a direct knock-on event; if the loading state in the channel is KWKWK, it is recorded as a soft knock-on event; and if the loading state in the channel is KWKK or KKWK, it is recorded as a mixed knock-on event.

Vibration power spectrum calculation: The vibrational power spectra of carbonyl groups in SF are obtained by recording their atomic velocities every 4 fs and then carrying out a Fourier transform on the autocorrelation function of the velocities. The vibrational power spectrum $I\left(\omega \right)$ is calculated as follows:(5)$$I(w)=\frac{2}{\pi }\int_{0}^{\infty }C(t)\cos(wt)dt$$
(6)$$C(t)=\langle v(t)\cdot v(0)\rangle /\langle {\left|v(0)\right|}^{2}\rangle$$
where *C*(*t*) is the autocorrelation function, *v*(*t*) is the atomic velocity, and *ω* is the vibrational frequency.

Hydrogen bond lifetime: Hydrogen bonds were determined according to geometric rules and are considered formed when the hydrogen bond donor–acceptor distance is less than 3.5 Å, and the angle formed by the hydrogen donor–acceptor is less than 30 degrees. The average hydrogen bonding lifetime is calculated by the probability distribution of the hydrogen bonding lifetime. We calculated this data by using the *gmx hbond* command in GROMACS.

## 4. Conclusions

In this paper, the influence of terahertz electromagnetic fields with different frequencies of 4 THz, 10 THz, 15 THz, and 20 THz on the ion permeation of the Kv1.2 voltage-gated potassium channel was studied by molecular dynamics simulation. The results show that the external electric field increases the ion flux to a different degree compared with without an external electric field. In particular, the external electric field with a frequency of 15 THz increases the ion flux by the most, up to 67.7%. The effect of an external electric field on the permeability of ion channels is analyzed as follows.

Firstly, by analyzing the influence of the applied electric field on the carbonyl groups of the SF binding sites, and potassium ions’ PMF in SF, we found that although the application of terahertz electric field has no strong resonance with the –C=O groups of the conservative TVGYG amino acid residues in the SF, it affects the stability of the electrostatic bond between potassium ions and carbonyl groups, bringing the change in ions’ potential at each binding site.

Secondly, by analyzing the influence of the applied electric field on the lifetime of the hydrogen bond formed at the SF entrance between water molecules and hydroxyl oxygen atoms on the side chain of 374THR, we found that the applied electric field could shorten the lifetime of the hydrogen bond. The probability of the “direct knock-on” mode can be increased, and the “soft knock-on” mode be decreased, resulting in the permeability of ion channels being increased. Therefore, the lifetime of the hydrogen bond becomes a main factor controlling ions permeation.

Compared to no applied field and an applied external electric field of 4 THz, 10 THz, and 20 THz frequencies, when the external electric field with 15 THz frequency is present, the lifetime of the hydrogen bond is reduced by 29%, the potential at the S_cav_ site is the highest, the probability of “direct knock-on” mode is increased by 45.5%, and the ion flux of the channel is improved by 67.7%. The result of our research supports the view that the “direct knock-on” mode is a faster permeation mode.

## Figures and Tables

**Figure 1 ijms-24-10271-f001:**
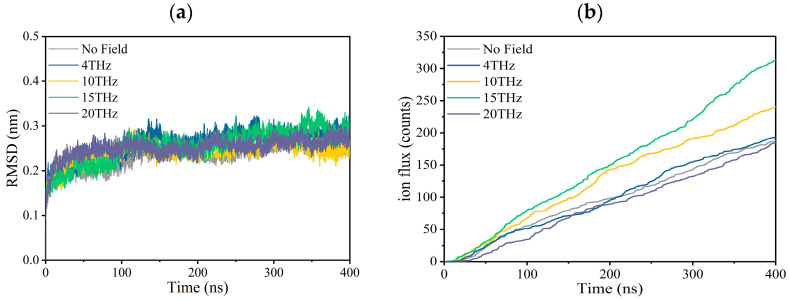
Effect of terahertz electric field on the property of the Kv1.2 channel. (**a**) Root Mean Square Deviation (RMSD) of the channel with or without an applied electric field. (**b**) Ion flux curves of potassium ion channels with or without an applied electric field.

**Figure 2 ijms-24-10271-f002:**
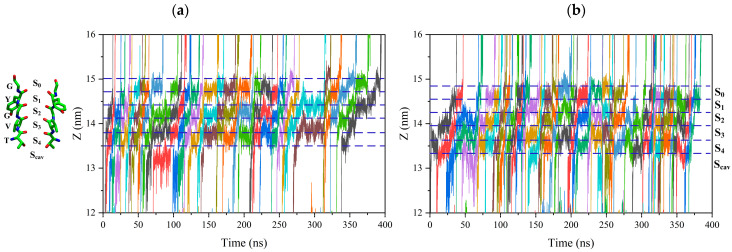
Trajectory of the ions through the SF region. Ion positions along the axis of the channel are shown for a 400 ns interval of one replica. Colored lines represent the positions of individual permeating ions (varying colors) vs. time. Dot lines correspond to the positions of the ion binding site, S_cav_, S_0_ to S_4_. (**a**) Trajectory of the ions through the SF region without an external electric field. (**b**) Trajectory of the ions through the SF region with a 15 THz external electric field.

**Figure 3 ijms-24-10271-f003:**
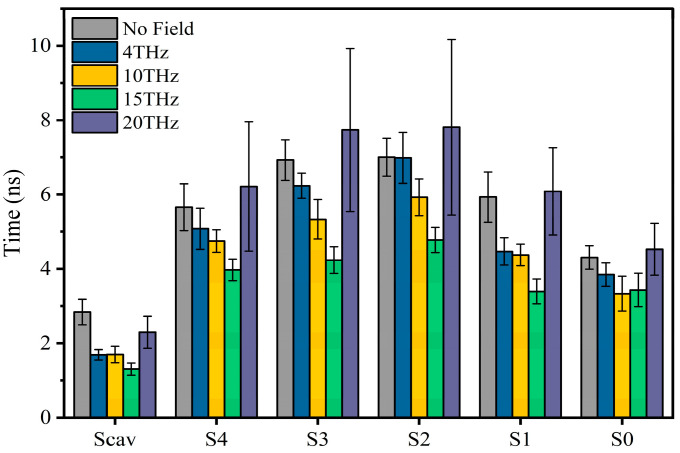
The vibration time of ions at SF binding sites without or with an applied electric field of different frequencies.

**Figure 4 ijms-24-10271-f004:**
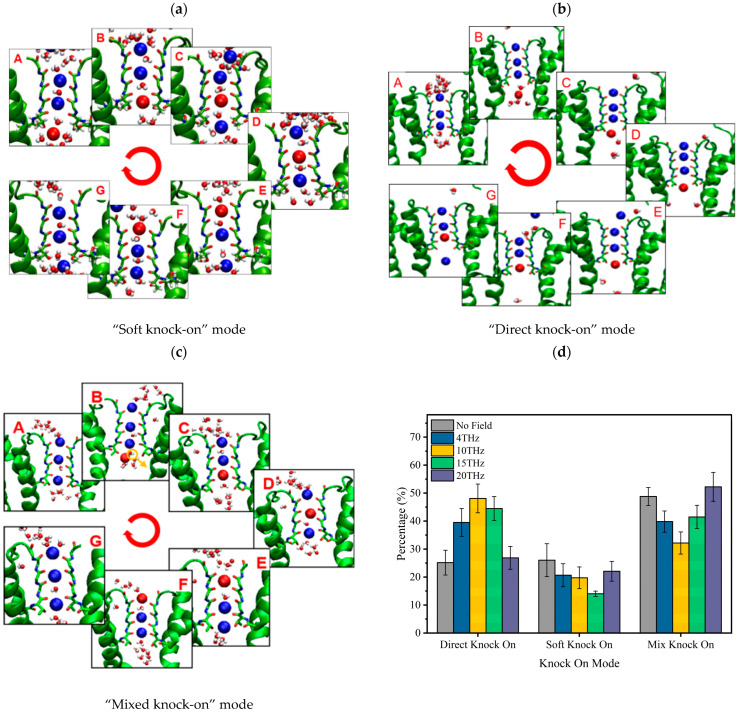
Molecular dynamics simulation of K^+^ permeation modes in SF. (**a**) “Soft knock-on” mode and sequence of events during K^+^ translocation. Snapshot for different loading sequences (A to G), descriptions of the occupied states of S_0_–S_4_ sites by potassium ion and water molecules (blue ball and red ball represent potassium ion and small white/red balls represent water molecules), and (**b**) “Direct knock-on” mode and sequence of events during K^+^ translocation. Snapshot for different loading sequences (A to G), descriptions of the occupied states of S_0_–S_4_ sites by potassium ion and vacancy (blue ball and red ball represent potassium ion and small balls of white and red represent water molecular), and (**c**) “Mixed knock on” mode and sequence of events during K^+^ translocation. Snapshot for different loading sequences (A to G), descriptions of the occupied states of S_0_–S_4_ sites by potassium ion, water molecular and vacancy (blue ball and red ball represent potassium ion and small balls of white and red represent water molecular), and (**d**) proportion of ion knock-on modes without and with an electric field of different frequencies.

**Figure 5 ijms-24-10271-f005:**
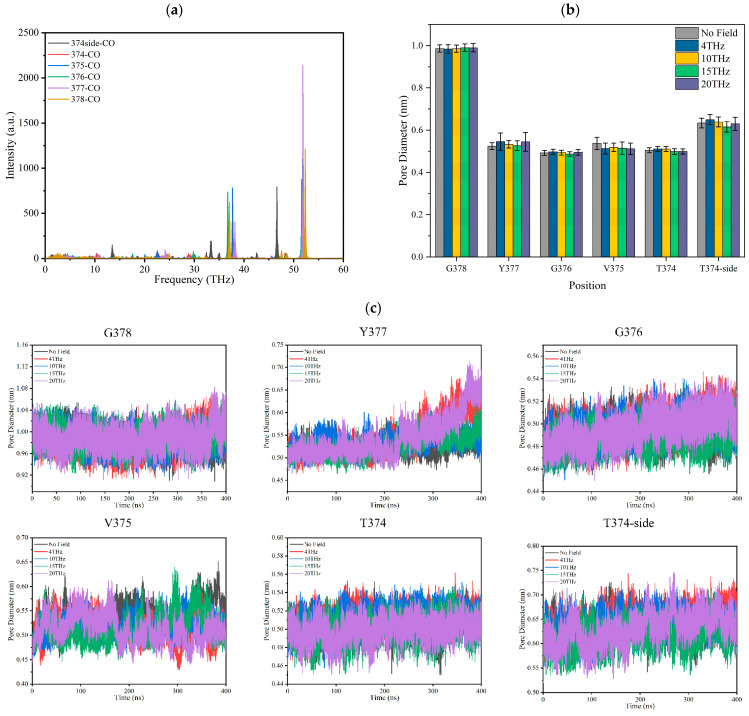
Effect of terahertz electric field on the –C=O of binding sites of the SF of Kv1.2 channel. (**a**) The infrared absorption spectrum of –C=O groups at sites S_0_–S_4_ in the SF region. (**b**) The average pore diameter of each binding site of SF under different conditions. (**c**) The distance over time between the carbonyl oxygen atom on each residue and the carbonyl oxygen atom on the symmetric residue in the SF.

**Figure 6 ijms-24-10271-f006:**
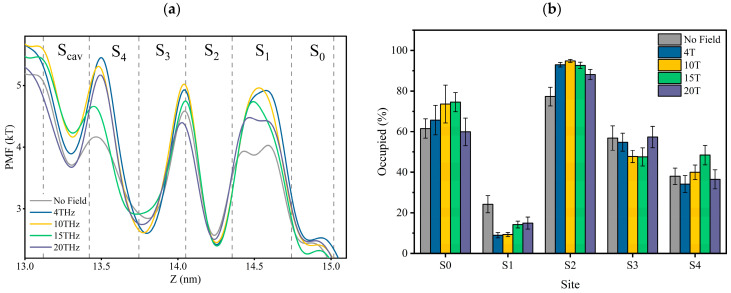
(**a**) Potential of mean force (PMF) for potassium ions in SF. PMF is the negative logarithmic of K^+^ ions’ density inside the SF. Vertical dotted lines are the positions of the carbonyl oxygen atom of SF at the binding site, and between the two dotted lines are the positions of the canonical potassium binding sites. (**b**) The occupancy rate of potassium ions at each site.

**Figure 7 ijms-24-10271-f007:**
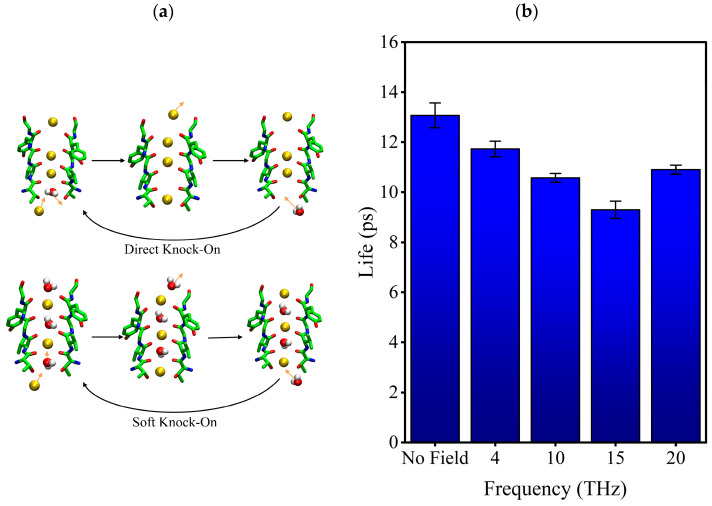
(**a**) The difference between the two transport modes of potassium ion. (**b**) Lifetime of hydrogen bond formed by water molecules and hydroxyl oxygen atoms on the side chain of 374THR.

**Figure 8 ijms-24-10271-f008:**
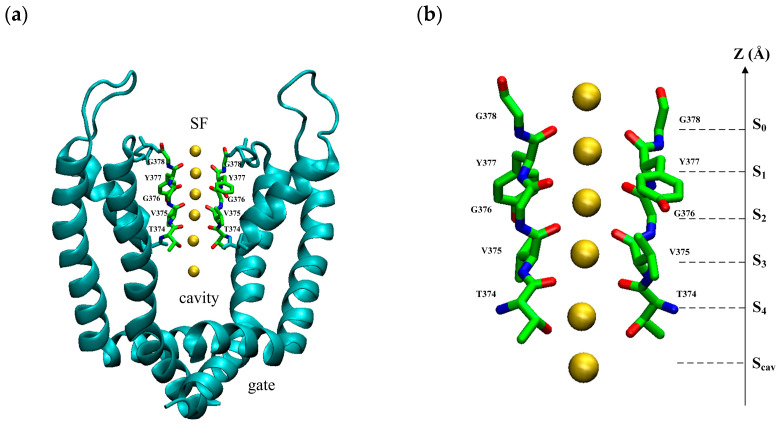
(**a**) The view of the pore domain of the homotetramer of Kv1.2 (PDB ID: 3lut) with an open intracellular gate near the bottom, cavity in the middle, and selective filter on the top. (**b**) The view of the selective filter with two opposite chains of the homotetramer. There are five main K^+^ binding sites (S_0_–S_4_) and an additional binding site in the water-filled central cavity (S_cav_). The yellow spheres are potassium ions.

**Figure 9 ijms-24-10271-f009:**
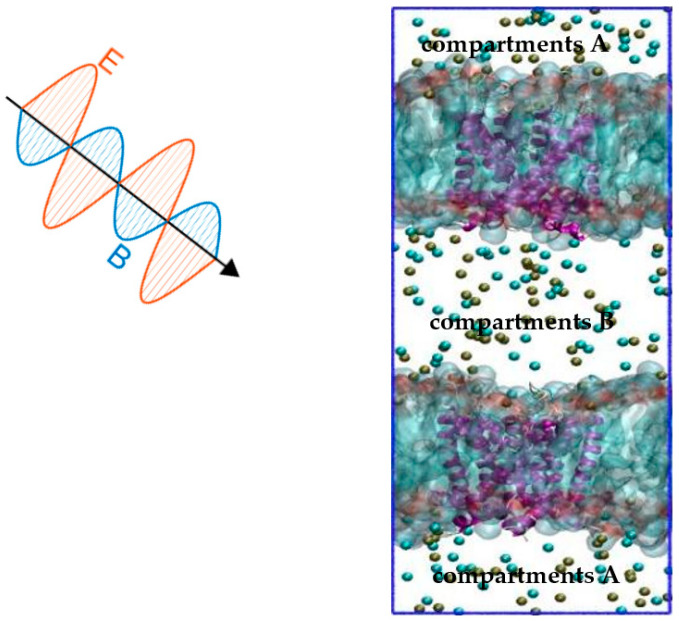
The CEF-IIMB combined model. The simulation system consists of two POPC membranes (light blue), each including open Kv1.2 (purple), which is surrounded by water and ions. The tan balls represent potassium ions, the cyan balls represent chloride ions, and water molecules are omitted for clarity. The transmembrane potential is generated by sustaining a small charge imbalance Δq between compartments A and B by the IIMB method. The CEF method is used to simulate the external THz-EM field.

**Table 1 ijms-24-10271-t001:** Electrical characteristics of potassium ion channels under terahertz electric fields with different frequencies.

Condition	Ion Flux (per/400 ns)	Ion Current (pA)	Ion Conductance (PS)
Without electric field	31 ± 3	12.5 ± 1.2	26.0 ± 3.1
4 THz	32 ± 4	12.9 ± 1.5	28.8 ± 3.4
10 THz	40 ± 4	16.0 ± 1.5	35.9 ± 3.6
15 THz	52 ± 4	20.9 ± 1.7	45.3 ± 4.0
20 THz	30 ± 6	12.3 ± 2.3	27.2 ± 4.9

## Data Availability

No new data were created or analyzed in this study. Data sharing is not applicable to this article.

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
