# Peer review of "Effect of Terahertz Electromagnetic Field on the Permeability of Potassium Channel Kv1.2"

_ijms, 2023, doi:10.3390/ijms241210271_

Round 1

Reviewer 1 Report

This study is an important extension of the findings in refs. 10 and 13 which showed that certain specific terahertz frequency EMF exposures act via their impact on the specificity filter structure of the voltage-gated potassium channel to increase potassium channel fluxes, rather than via impacts on the voltage sensor.  The earlier refs showed similar findings with regard to the L-type voltage-gated calcium channels.  The only change that I suggest is that the authors in the abstract change the word improved to activated.

Reviewer 2 Report

The manuscript "Effect of terahertz electromagnetic field on the permeability of potassium channel Kv1.2 " by Ding et al. reports a simulation study of potassium ion passage through the Kv1.2 channel under different conditions, as mentioned in the title. The authors find highest ion conductivity for a 15 THz field, the second highest frequency analysed in the present work and an increase of a "hard knock" mode of ion transitions under that condition. Based on theri analysis they rationalise that it is not C=O vibration resonances that change the ion transport (speed and mode) but rather hydrogen bonds between water molecules and T374 at the entrance of the channel that are affected by the applied fields.

The work is interesting as it sheds light on ion transport under different conditions in atomic details. There are, however, an number of major concerns that need to be addresses before publication can be considered. 

The results cannot be properly evaluated without 1) sufficient description of the analyses performed and 2) suitable error estimates.

Regarding 1):

How have the residence times been calculated? Is this by counting how long an ion stays close (by a distance criterion) to a certain site or Z-value? This must be explained in the methods section.

How have the sequences of events be determined? That is, have potassium ions and water molecules and "vacancies" been traced? If so, how?

How has the vibrational spectrum been calculated? The authors should be more verbatim here (in the methods section) than providing only one reference. 

How have the life times of hydrogen-bonds been calculated?

Regarding 2):

The authors performed "six production simulations of 400ns" for each condition. Are the reported values averages from these six simulations? These many production runs per condition allow an error estimation by the standard error from the mean, the mean taken as the average over the quantities calculated from the six production runs. However, no error estimates are rported. It is therefore impossible to evaluate the statistical significance of differences between the different conditions. Figure 7B is the only exception here that shows error bars.

Figure 3 reports time series of RMSD and ion flux. These cannot be averages (since this is not meaningful). Which of the six runs were used here?

Other major concerns:

The authors find an increase in ion flux, and correspondingly in ion current or ion conductance, with increasing  frequency of the terahertz electric field. However, this holds only up to 15 Thz. The ion flux for 20 THz is comparable to that without or with a low-frequency field. How can this be explained?

The statement that "the 'soft-knock on' mode is not an effective permeation mode for potassium ion channels." (line 256-257) due to it not being prevalent at 15 Thz cannot be generalised in that way.  It has lower abundance in the simulations at 15 Thz (at least that is what the data suggest), but, according to Figure 6, without terahertz field, it is compabrable to the "direct knock-on mode" . The statement has to be rephrased.

Figure 7a) shows some intensity for "374side-CO" (black curve), whatever this refers to. Even if that is not a strong absorption peak (the authors state there are none), it is nevertheless non-neglible. However, without knowing how the intensities have been calculated it is meaningless to speculate about them. The legend of Figure7A must be improved or better explained in the caption. Which fields are employed here?

The distance between the carbonyl oxygen atoms of the Y377 residues changes over time under 20THz and 4 Thz, and perhaps also somewaht with 15THz as far as can be seen from Figure 7C. And also the distances between the G376 residues increase under 20 THz.Yet, the authors do not find any change in pore diameter. How can this be explained?

Moreover, it is interesting too not that the increase in Y377-Y377 distance is comparable for 4 and 20 THz, the two conditions which show the lowest ion flux (apart from no field). How can this be explained? HAving more space (larger diameter) cannot be the explanation since the residence times at the binding sites are also highest under these conditions with low ion flux.

It is unclear what is meant by "very close" in "the stably bound ions (potential minima) in S4 is very close to that of S3," (line 308). Do the authors refer to the Z-value? That does not necessarily mean small barriers and is also not necessary to assume when the PMF with all barriers is provided. Perhaps this needs to be simply rephrased since the authors discuss the height of the barriers shown in the PMF in the following.

The location of the barriers are also different in case of the 20 THz field. This can be a consequence of the Y377-distances having increased ind with that the definition of the ion binding site along Z as well. This must be discussed.

The authors discuss the S4/Sc barrier being minimlal for the 15 Thz field, for which condition they found maximal ion flux. But the PMF in Figure 8(s) shows an even lower barrier  for the simulations without field. It is also this no-field condition which shows the lowest S0/S1 barrier. This seems to be a contradiction to the authors' claim. What should be taken into accountthough, is the  very similar barriers beyond(or rather before) the Sc binding site, which are the highest in the entire PMF. So the entrance to the Sc site (from below) appears to be the rate-determining step in ion passage (and not S4/Sc). This must be discussed.

The authors claim "When the water molecular become free, there is no energy barrier to hinder it return to the cavity, so it's easier to get back into the cavity than it is to get into the S4"

(line 358). Did they calculate PMF for water movements? What about a barrier or no such barrier in the other direction? The free water molecule could move just as well towards S4.

It would als be insightful to analyse the probabilities and life times for water molecules at the different binding sites so as to get a better basis for dicussing the prevalenece (or not) of the differnt mechanisms.

The authors should use the same colour code for the different conditions throughout the manuscript to increase their recognisability in the different figures. The colour codes used in Figure 3 are different from those in Figures 5, 6 and 7, but then used again in Figure 7 and having a (though different) version of green and purple for different conditions. And Figure 8 has yet another colour code. That is confusing. 

Other minor points:

There is a space missing between the values and THz throughout the manuscript. And occasionally there is "xxTHz terahertz" written instead of only "xx THz" (xx being some value here).

I reckon " the potassium hydrate ion" (line 152) is a hydrated potassium ion. This must be corrected.

The manuscript needs some language editing. to give a few examples:

Line 65-66 "Zhang et al. demonstrated that 0.1THz terahertz electric field can increase the

channel current of KcsA ion channel by 4 times through molecular dynamics simulation" should be changed to "Zhang et al. demonstrated  through molecular dynamics simulation that 0.1THz terahertz electric field can increase the channel current of KcsA ion channel by 4 times" since it could be misunderstood of the chnnel performing moecular dynamics simulations.

line 72: change "electromagnetic fields may have an interact" to "electromagnetic fields may interact"

line 77 : "much researches need to do" "much research needs to be done"

In line 96, ion binding sites are introduced as "Sc,S0, S1, S2, S3, S4,S0 " which would give the same label, S0, to two binding sites.  Moreover, Figure 1 shows only binding sites S0 to S4, but no Sc. 

There is also a typo in the legend of Figure 8(b).

English editing is required.

Round 2

Reviewer 2 Report

The authors have addressed all my concerns and this significantly improved manuscript is now acceptable for publication.

Spell checking and minor proof reading required .